# Steady Moderate Exercise Confers Resilience Against Neurodegeneration and Neuroinflammation in a Mouse Model of Parkinson’s Disease

**DOI:** 10.3390/ijms26031146

**Published:** 2025-01-28

**Authors:** Ewelina Palasz, Anna Gasiorowska-Bien, Patrycja Drapich, Wiktor Niewiadomski, Grazyna Niewiadomska

**Affiliations:** 1Department of Cellular Signalling, Mossakowski Medical Research Institute, Polish Academy of Sciences, 02-106 Warsaw, Poland; epalasz@imdik.pan.pl; 2Clinical and Research Department of Applied Physiology, Mossakowski Medical Research Institute, Polish Academy of Sciences, 02-106 Warsaw, Poland; agasiorowska@imdik.pan.pl (A.G.-B.); pdrapich@imdik.pan.pl (P.D.); wniewiadomski@imdik.pan.pl (W.N.); 3Nencki Institute of Experimental Biology, Polish Academy of Sciences, 02-093 Warsaw, Poland

**Keywords:** Parkinson’s disease, exercise intensity, MPTP animal model, inflammation, neuroprotection, neurotrophins, PD therapy

## Abstract

Intensive aerobic exercise slows the progression of movement disorders in Parkinson’s disease (PD) and is therefore recommended as an important component of treatment for PD patients. Studies in animal models of PD have shown that vigorous exercise has neuroprotective effects, and emerging evidence suggests that it may be a disease-modifying treatment in humans. However, many people with PD may not be able to participate in vigorous exercise because of multiple medical conditions that severely limit their physical activity. In this study, we have shown that chronic MPTP treatment in sedentary mice resulted in loss of dopaminergic neurons in the SNpc, decreased levels of neurotrophins, BDNF and GDNF, and increased levels of inflammatory markers and pro-inflammatory changes in immunocompetent cells. Moderate exercise, initiated both before and after chronic MPTP treatment, significantly attenuated the loss of dopaminergic neurons and increased BDNF and GDNF levels even above those in sedentary control mice. No signs of inflammation were observed in MPTP-treated mice, either when training began before or after MPTP treatment. Training induced beneficial changes in the dopaminergic system, increased levels of neurotrophins and suppression of inflammation were similar for both steady moderate (present data) and intense training (our previously published data). This suggests that there is a kind of saturation when the percentage of rescued dopaminergic neurons reaches the highest possible value, and therefore further increases in exercise intensity do not enhance neuroprotection. In conclusion, our present results compared with the previous data show that increasing exercise intensity beyond the level used in this study does not increase the neuroprotective effect of aerobic training in a mouse model of Parkinson’s disease.

## 1. Introduction

Parkinson’s disease (PD) is a degenerative disorder of the central nervous system whose primary symptom is movement disorders resulting from the degeneration of dopaminergic neurons in the nigrostriatal pathway. The etiology of the disease is not fully understood due to the complex underlying mechanisms. Existing therapies only alleviate the symptoms of the disease and become less effective as the disease progresses [1]. Therefore, it is extremely important to search for new therapies to support drug treatment. Increasing attention is being paid to the effects of exercise on improving or maintaining normal brain function. Studies in healthy elderly populations without central nervous system disorders have shown that regular aerobic activity induces changes related to neuroplasticity, including synaptogenesis, increased glucose consumption, angiogenesis, and neurogenesis. Exercise improves brain function by inhibiting inflammation, reducing oxidative stress, stabilizing calcium homeostasis, and stimulating the release of endogenous neurotrophins [2]. The importance of training and exercise is currently being studied extensively in animal models of PD. Many of these studies suggest that exercise may have a neuroprotective effect, slowing or even stopping neurodegenerative processes and restoring disrupted signaling pathways [3]. Based on the findings in humans and animal models of PD, physical activity has been found to improve motor behavior patterns and enhance angiogenesis, synaptogenesis, and neurogenesis in the brain. It also increases levels of neurotrophic factors and reduces the severity of inflammatory processes [4].

Physical activity has been shown to have beneficial effects on both motor and non-motor symptoms in PD. The mechanisms involved in these effects are neuroendocrine and neurotransmitter regulation, enhancement of neuronal insulin signaling, upregulation of neurotrophins and the myokine irisin, attenuation of inflammatory and oxidative responses, enhancement of pro-survival and anti-apoptosis effects, and optimization of the autophagy process. Exercise has also been shown to have beneficial effects on a number of intracellular signaling pathways involving Ca^2+^, cyclic AMP-responsive element-binding protein (CREB), nuclear factor kappa B (NF-κΒ) and peroxisome proliferator-activated receptor gamma coactivator 1-alpha (PGC-1α), protein chaperones, neurotrophic factors, DNA repair proteins, mitochondrial biogenesis (for a review of this, see [5,6]). Therefore, the question of whether exercise, particularly vigorous exercise, can be used as a disease-modifying therapy in people with PD remains central to clinical management. Uhrbrand et al. [7] and Gamborg et al. [8] reviewed randomized controlled trials assessing the functional effects of different types of vigorous exercise in PD patients. All types of exercise were found to be safe and feasible, did not worsen PD symptoms, and were recognized as beneficial adjunctive rehabilitation strategies for this disease. Schenkman et al. [9] compared the functional effects of intensive treadmill exercise to those of moderate-intensity treadmill exercise and found that only the former significantly slowed the time-dependent decline in motor function observed in PD. Based on data from a number of studies, Alberts and Rosenfeldt [10] published an article recommending the inclusion of vigorous regular aerobic exercise in the treatment of patients with PD. However, Parkinson’s disease may discourage or even prevent participation in intensive training due to muscle weakness, which is a primary symptom in PD [11], sympathetic denervation of the heart, which reduces cardiac contractility [12], debilitating and persistent fatigue, apathy, which is often reported in PD, and increased fatigability, which may be caused by reduced cardiorespiratory fitness [13]. In light of the above, it is important to consider the following questions: What is the role of exercise intensity in the mechanism that modifies the effect of exercise on the course of PD?

It is clear that one of the most useful, if imperfect, ways of investigating these questions at present is in animal models of PD. A considerable number of studies have shown that exercise has neuroprotective effects on dopaminergic neurons in rodent models of PD [14], and the results have been summarized in several reviews [15,16]. However, the effects of the type of exercise and its dose, in terms of either intensity or volume, have not been systematically investigated. An example of such a study is the work of Gerecke et al. [17], who investigated the neuroprotection against 1-Methyl-4-phenyl-1,2,3,6-tetrahydropyridine (MPTP) intoxication provided by exercise performed prior to neurotoxin administration. The authors found that there is a certain distance of daily running in mice on a running wheel that completely prevents neuronal loss. However, 2/3 of this distance provides only partial protection, and 1/3 of the distance run has no neuroprotective effect. We are not aware of similar studies of exercise-induced neuroprotection when exercise was performed during or after MPTP application.

Our previous study [18] looked at the time of the onset of exercise (before or after intoxication) required for the induction of neuroprotection in a mouse model of PD induced by MPTP. A clear effect of exercise on neuronal survival, neuroinflammation, and the expression of a number of growth factors was observed after high-intensity training. As the role of exercise intensity as a variable that may be relevant to the disease-modifying effects of exercise in PD has been highlighted in human studies, we decided to present our results of what exercise intensity is necessary to induce neuroprotection in an environmental model of PD. The current study analyzes the neuroprotective effect of less intense exercise. To address the question of whether the neuroprotective effect of physical exercise depends on the intensity of the training, the current study was conducted using exercise of moderate intensity. The results of moderately trained mice are compared to an untreated and untrained control group and an MPTP-treated and untreated group, which are the same as those in both previous and current studies. As a result, the only difference between the previously published data and those of the present study was the intensity of the exercise. By comparing the results of this and the previous experiment, it is possible to assess the effect of changing just one variable—exercise intensity.

## 2. Results

### 2.1. Treadmill Exercise Neuroprotects the Nigrostriatal Dopaminergic System in a Chronic MPTP Mouse Model of Parkinson’s Disease

MPTP is a well-known neurotoxin responsible for the extensive and relatively selective destruction of dopaminergic neurons in the nigrostriatal pathway [19,20]. To assess changes in dopaminergic projections in the nigrostriatal pathway in a mouse model of Parkinsonism, immunohistochemical staining against tyrosine hydroxylase (or tyrosine 3-monooxygenase; EC 1.14.16.2; TH) and a computer-assisted densitometric analysis of TH-ir neurons in the SNpc and the target dopaminergic projection in the striatum were performed. TH is an enzyme of the hydroxylase group, responsible for converting the amino acid L-tyrosine to dihydroxyphenylalanine (DOPA). This is an important step in the biosynthesis of dopamine in neuronal cells. This enzyme is highly substrate-specific, and TH is therefore considered a useful marker for dopaminergic neurons.

A significant decrease in staining intensity was observed in the SNpc as well as in the ST, which receives dopaminergic projections from the SNpc, after MPTP administration in untrained mice (MPTP group) (Figure 1, microphotographs on the left and right vertical panels). There was a significant reduction of 66% in the number of TH-ir neurons in the SNpc in untrained MPTP mice compared to control mice not receiving neurotoxin (group C). The application of moderate-intensity treadmill training in MPTP-treated mice resulted in a lower loss of TH-positive neurons in the SNpc. The reduction in TH-ir neurons reached 27% in the MPTP-TE group and 18% in the MPTP-TL group compared to the untrained control group (group C). It should be emphasized that this difference in the number of TH-ir neurons was not statistically significant, which may indicate a beneficial neuroprotective effect of exercise. Unexpectedly, the number of TH-ir neurons was lower (by 13%) in the trained control group (C-T) compared to the non-exercise group C, but this difference was not statistically significant. Therefore, the differences in the number of TH-ir neurons in the MPTP-TE and MPTP-TL groups were less pronounced compared to the C-T group, by 16% and 6%, respectively. Again, these differences were not statistically significant. The analysis also showed that the time of training onset did not significantly affect the number of TH-ir cells in the SNpc in the MPTP-TE and MPTP-TL groups (Figure 1A).

In the striatum, TH expression was assessed by the OD and %Area. A significant reduction of 53% in OD and 73% in %Area was observed in the group of untrained MPTP-treated mice (MPTP) compared to untreated and untrained control mice (C). Treadmill training applied to untreated control mice (C-T) increased OD (by 18%) and %Area (by 8%) compared to untreated controls (C), but the difference was not significant. The application of training to MPTP-treated mice resulted in the preservation of TH staining intensity. Early-onset training (MPTP-TE) resulted in a 6% increase in OD and a 3% increase in %Area, while late-onset training (MPTP-TL) resulted in a 1% decrease in OD and a 2% decrease in %Area compared to group C. These differences were not statistically significant. There were also no significant differences between the MPTP groups treated with early- or late-onset training. Thus, in the striatum, similar to the SNpc, training prevented the decrease in TH staining induced by MPTP (Figure 1B,C).

DRD2 dopamine receptors play a key role in the function of direct and indirect pathways in the striatum, which connect to peripheral output systems to influence motor behavior. The direct pathway is associated with movement activation, while the indirect pathway is associated with movement inhibition. The presence of only one type of dopamine receptor in a single brain region is rare, and one such structure is the dorsal striatum in mice, which has only DRD2 dopamine receptors [21]. Therefore, the assessment of DRD2 receptor expression levels may be a good marker for abnormalities in the projection of dopaminergic neurons in the nigrostriatal pathway.

In our study, DRD2 immunohistochemical staining in the striatum showed the strongest intensity in untrained MPTP-treated mice (MPTP group) and was significantly stronger than in all other groups (Figure 2A). In the MPTP group, DRD2 immunoreactivity was statistically significantly higher by 64% in OD and 37% in %Area compared to group C. Training alone did not significantly affect the intensity of DRD2 immunoreactivity; there were slight decreases in OD and %Area (3% and 2%, respectively) in the C-T group compared to group C. Training in MPTP mice prevented the increase in DRD2-ir staining intensity. There were no significant differences between untreated control groups (C and C-T) and trained MPTP groups (MPTP-TE and MPTP-TL). There was only a 2% increase in OD and a 2% increase in %Area in the MPTP-TE group compared to group C, and there was a 5% increase in OD and a 4% increase in %Area compared to the C-T group. Late-onset training (MPTP-TL group) resulted in a 15% increase in OD and a 13% increase in %Area compared to group C as well as a 19% increase in OD and a 14% increase in %Area compared to the C-T group (Figure 2B,C).

### 2.2. Effect of Treadmill Exercise and the Administration of MPTP on the Levels of Endogenous Neurotrophic Factors

Studies suggest the involvement of trophic factors such as BDNF and GDNF, which have a protective effect on dopaminergic neurons. It is noteworthy that physical activity may influence neuroplasticity mechanisms in PD related to neurotransmission in the dopaminergic and glutamatergic systems [22,23,24]. Therefore, the effects of physical training and MPTP administration on BDNF and GDNF levels were investigated by immunohistochemistry in the SNpc and striatum and by an ELISA assay in the midbrain and striatum.

Anti-BDNF staining in the SNpc showed the strongest intensity in the MPTP-trained groups, while the weakest intensity was present in the MPTP-untrained group (Figure 3, left vertical panel). The ELISA assay showed that in the midbrain, BDNF levels were 14% lower in untrained MPTP-treated mice compared to untreated mice (group C), but this difference was not significant. Exercise alone (C-T vs. group C) resulted in a 22% increase in the BDNF concentration, which was not significant. Physical training combined with MPTP administration significantly increased the upregulation of BDNF by 81% in MPTP-TE and 74% in MPTP-TL compared to the untrained control (group C) and by 49% in MPTP-TE and 43% in MPTP-TL compared to the C-T group. However, the timing of training did not matter as both early- and late-onset training resulted in similar increases in BDNF concentrations in the MPTP-TE and MPTP-TL groups (Figure 3B).

Similar to the SNpc, anti-BDNF staining in the striatum was strongest in the trained MPTP groups and weakest in the untrained MPTP group (Figure 3, right vertical panel). The ELISA assay showed that BDNF levels in the striatum of untrained MPTP mice (MPTM group) were similar to those observed in untreated mice (group C). When training was applied to animals not treated with neurotoxin (C-T group), the concentration of BDNF increased significantly by 27% compared to the untrained group (group C). The application of training to MPTP groups also caused a significant increase in BDNF concentration, reaching 24% and 27% in MPTP-TE and MPTP-TL, respectively, compared to untrained control mice (group C). In contrast to the midbrain, MPTP treatment did not potentiate the training-induced upregulation of BDNF levels in the striatum. BDNF levels were similar in the C-T, MPTP-TE, and MPTP-TL groups (Figure 3B).

Anti-GDNF staining in the midbrain was strongest in the MPTP-TE and MPTP-TL groups and weakest in the untrained MPTP group (Figure 4, left vertical panel). The ELISA test showed that the midbrain GDNF level was 4% lower in the untrained MPTP group compared to untreated mice (group C). This difference was not statistically significant. However, when training was applied to the control group (C-T group), the GDNF level was significantly higher than in the untrained MPTP group. MPTP treatment combined with training increased the GDNF level, which was similar to the level in the C-T group and significantly higher than that in the MPTP group without training. The timing of training application (early- vs. late-onset of training) did not result in differences between the MPTP-TE and MPTP-TL groups with regard to GDNF levels (Figure 4A).

As in the midbrain, anti-GDNF staining in the striatum was most intense in the MPTP-TE and MPTP-TL groups and weakest in untrained MPTP mice (group C) (Figure 4, right vertical panel). Compared to the midbrain, MPTP administration caused a significantly greater decrease in GDNF levels in the striatum in untrained MPTP mice (by 4% in the midbrain vs. 13% in the striatum) (Figure 4A,B), and the decrease in GDNF levels was significant compared to group C. The pro-trophic effect of exercise was enhanced in mice receiving MPTP. Training significantly increased GDNF levels by 9% in C-T groups, 28% in MPTP-TE, and 22% in MPTP-TL compared to untrained control mice (C). Consequently, GDNF levels were also significantly higher in the MPTP-TE group and the MPTP-TL group, by 17% and 12%, respectively, compared to treadmill-exercising control mice (C-T). Similar to the midbrain, the timing of training application did not result in differences between the MPTP-TE and MPTP-TL groups with regard to GDNF levels (Figure 4B).

### 2.3. Pro-Inflammatory Activation of Glial Cells After the Administration of MPTP Is Attenuated After Physical Training

Chronic inflammation plays an important role in the development of PD. On the other hand, it is not yet clear whether inflammation is a primary or secondary phenomenon, i.e., a consequence of neuronal death [25]. Characteristic features of neuroinflammation include activated microglia and reactive astrocytes, which are known to produce cytokines, chemokines, prostaglandins, protein complement cascades, and reactive oxygen and nitrogen species [26]. Our present study involved an immunohistochemical analysis of GFAP, Iba-1b, and integrin CD11b in the SNpc and striatum to assess the neuroinflammation evoked by MPTP and the potential neuroprotective influence of applied training on astrocytes and microglia.

Significant staining intensity for GFAP, Iba-1b, and integrin CD11b in the SNpc and ST was observed in sedentary MPTP mice (Figure 5A,B). Immunofluorescence staining for GFAP showed increased recruitment of astrocytes in the SNpc and ST of the sedentary MPTP mice compared to all other groups. In addition, microglial recruitment was observed in mice with induced Parkinsonism. Immunofluorescence for CD11b and Iba-1b, markers of microglia, also showed higher intensity in the SNpc and ST in MPTP-untrained mice compared to results obtained in controls (C and C-T groups) and both MPTP-treated and treadmill-trained groups (MPTP-TE and MPTP-TL groups) (Figure 5A,B). In addition, in the MPTP-treated group, higher magnification of anti-Iba-1b and anti-CD11b staining in both the SNpc and ST indicated the morphological transformation of resting microglia into activated cells resembling pro-inflammatory amoeboid phagocytic cells (Figure 5A,B insertions).

In contrast, we found that exercise appeared to prevent the development of inflammatory processes mediated by astrocytes and microglia cells, despite MPTP intoxication. Microscopic analysis revealed a significantly lower intensity of staining against GFAP in the SNpc and ST in the MPTP-TE and MPTP-TL treadmill-trained groups of mice compared to the untrained MPTP group, comparable to the staining in the control groups, especially when compared to the C-T group (Figure 5A,B). Iba-1b-ir and CD11b-ir in the SNpc and ST of the MPTP-TE and MPTP-TL groups were similar to those observed in the C and C-T groups, and the morphology of the microglial cells resembled that of non-activated microglial cells (Figure 5A,B, insertions in the anti-Iba-1b and anti-CD11b panels).

High levels of inflammatory markers and pro-inflammatory changes in the morphology of immunocompetent cells were observed only in MPTP-treated sedentary mice. No signs of inflammation were seen in MPTP-treated mice when training started either before or after MPTP administration. Thus, the persistent inflammation induced by MPTP was abolished by physical exercise.

## 3. Discussion

The results presented above confirm observations that physical activity counteracts damage to midbrain dopaminergic neurons in an animal model of PD [17,18,27] and that this effect is accompanied by an increase in GDNF and BDNF levels in the substantia nigra pars compacta and striatum [28,29,30] and a reduction in the development of inflammation in dopaminergic structures [31,32,33,34,35]. Our data are of clinical relevance as they provide important evidence that exercise training may be a beneficial form of therapy in patients with Parkinson’s syndrome, even if it is not characterized by high exercise intensity.

### 3.1. Steady Moderate Physical Training Attenuates the Loss of Dopaminergic Neurons Induced by the Chronic Administration of MPTP

Our study showed that a steady moderate 10-week exercise regimen applied immediately after chronic 5-week MPTP treatment significantly attenuated the loss of dopaminergic neurons in the SNpc and normalized DRD2 receptor levels and the density of dopaminergic innervation in the striatum, similar to the same 10-week exercise regimen initiated 1 week prior to such MPTP treatment in mice.

The neuroprotective effects of exercise on the nigrostriatal system have been demonstrated in animal models of PD [36]. Numerous studies in these models have confirmed that high physical activity attenuates the damaging effects of the dopaminergic neurotoxins 6-OHDA and MPTP, has neuroprotective effects on dopaminergic neurons in the SNpc, and normalizes the integrity of dopaminergic terminals in the striatum. An analysis of the number of dopaminergic neurons in the midbrain confirmed the neuroprotective effect of exercise in most [17,37,38], but not all, studies [39,40]. The neuroprotective effect was found to be dose-, duration-, and intensity-dependent, with each of these elements having an effect on neurochemical measures, neuronal counts, and motor symptoms of Parkinsonism [17,37]. In contrast, our previously published results [18] demonstrated that the training, which began before the administration of the neurotoxin and was applied after the end of the intoxication, (1) almost completely preserved the number of dopaminergic neurons in the SNpc and ventral tegmental area (VTA), (2) increased the BDNF level in the midbrain and the GDNF level in the striatum to a similar degree, and (3) entirely prevented inflammatory responses evoked by chronic MPTP treatment.

### 3.2. Neurotrophins Are Involved in the Neuroprotective Effects of Steady Moderate Exercise in a Mouse Model of Parkinson’s Disease

Steady moderate physical training for 10 weeks increased BDNF in the midbrain and striatum in trained non-MPTP-treated mice. Chronic MPTP treatment in sedentary mice decreased the levels of this neurotrophin. Training that started both before and after chronic MPTP treatment increased BDNF levels not only above those of non-MPTP-treated sedentary mice but also above those found in the midbrain of trained non-MPTP-treated mice.

For GDNF levels, the greatest differences induced by training were observed in the striatum. Training increased the level of GDNF in non-MPTP-treated mice above the level observed in sedentary non-MPTP-treated mice. Chronic MPTP treatment in sedentary mice decreased the levels of this neurotrophin. Surprisingly, in the striatum, MPTP treatment potentiated the effect of training as GDNF levels in trained MPTP-treated mice increased even above those observed in trained non-MPTP-treated mice. However, in the midbrain, MPTP treatment did not potentiate the effect of training on GDNF levels, but training in MPTP groups maintained midbrain GDNF levels similar to those observed in the trained non-MPTP-treated group.

The available literature supports the central role of the neurotrophic factors BDNF and GDNF in exercise-induced neuroprotection. In addition to the fact that exercise increases the levels of neurotrophic factors, it is also known that supplementation with exogenous BDNF or GDNF alone is sufficient to prevent the loss of dopaminergic neurons following the damaging effects of toxins, as demonstrated both in vitro and in animal model studies [41,42,43].

In contrast, the reduced synthesis of GDNF or BDNF after the knock-out of their genes or silencing of their expression by antisense oligonucleotides results in a progressive loss of TH-positive neurons in the SNpc of mice and increases their sensitivity to MPTP [44,45]. Exercise also fails to protect against MPTP-induced neurotoxicity in mice heterozygous for the BDNF gene (BDNF^+/−^) [17,24]. It has also been shown that the application of a TrkB receptor antagonist causes physical activity to lose its protective effect on dopaminergic neurons [43]. These results suggest that physical activity may reduce the sensitivity of dopaminergic neurons to toxins via the activation of signaling cascades triggered by the increased availability of BDNF and GDNF. The increased number of immunohistochemically labeled GDNF cells observed in exercising mice may indicate exercise-induced mobilization of glial cells and the activation of anti-inflammatory processes.

### 3.3. Anti-Inflammatory Effect of Steady Moderate Exercise in a Mouse Model of Parkinson’s Disease

Immunohistochemical detection of increased levels of GFAP, Iba-1b, and CD11b may indicate that both astroglia and microglia are involved in the inflammatory response or pathological changes in nigrostriatal structures in mice receiving MPTP. High levels of inflammatory markers and pro-inflammatory changes in immunocompetent cells were observed only in MPTP-treated sedentary mice. The signs of inflammation were still present four weeks after the end of MPTP treatment, but they were absent in the early-onset training group examined at the same time point, i.e., also four weeks after the end of neurotoxin treatment. Thus, the persistent inflammation induced by MPTP was abolished by exercise.

Iba-1b and CD11b proteins are among the markers of immune cells and, in the context of microglia, are specific markers for these cells [46]. Increased expression of Iba-1b and CD11b indicates that microglia may be activated in a given tissue area, which may be indicative of inflammation, immune response, or neuroinflammatory processes, as both proteins are associated with phagocytic functions and inflammatory cell activity [47].

CD11b, together with other transmembrane and surface proteins, is the first line of defense against pathogens. It is involved in adhesion processes and the uptake of complement-coated molecules [48]. It has been reported that increased CD11b expression in various neuroinflammatory diseases corresponds to the severity of microglial activation [49]. Morphologically, microglial activation is associated with intense branching and cytoskeletal rearrangement, with changes in shape and motility correlating with an increased expression of CD11b [50]. During this activation process, the cytoplasmic domain of CD11b is thought to interact increasingly with cytoskeletal proteins [51].

The most commonly used protein marker of microglial activation is also an elevated level of Iba1. This is a member of the calcium-binding protein group. Iba1 is an intracellular protein, and its functions are related to the reorganization of the microglial cytoskeleton and the support of the phagocytosis process. The latter is possible thanks to its ability to bind actin molecules [52]. This protein is one of the most widely studied in inflammatory processes due to its conservative amino acid sequence and stability of antigenic epitopes across species, including humans [53].

Current knowledge of the mechanisms involved in the anti-inflammatory effects of exercise training in PD is based on data obtained in animal models [54], and these studies are not very numerous. The results obtained by Sconce et al. [55] show that animals treated with increasing doses of MPTP for 4 weeks had significantly increased levels of GFAP in the SNpc, suggesting that astrocytosis processes were activated. Interestingly, the MPTP-treated and exercising groups also had elevated GFAP levels compared to the control group, but this was significantly lower than the group of mice treated with MPTP but not exercising. This suggests that exercise may suppress the inflammatory response in the brain, including the attenuation of astrocytosis. Another study, also in a mouse model using the neurotoxin 6-OHDA [32], showed that both aerobic treadmill training and resistance training have neuroprotective effects, probably by stimulating sirtuin 1 activity. This can regulate both mitochondrial function and neuroinflammatory processes through the deacetylation of NF-κΒ. Both forms of training also reduced levels of nitric oxide (NO), TNF-α (tumor necrosis factor alfa), IFN-γ (interferon gamma, a type 2 interferon produced by leukocytes), IL-1β (interleukin beta 1), and TGF-β1 (transforming growth factor beta 1) in mice receiving 6-OHDA. The regulation of NO levels may also be the mechanism by which exercise training affects the magnitude of the inflammatory response induced by 6-OHDA administration.

It is reasonable to suggest that the exercise-induced reduction in the inflammatory response may be the result of a direct effect on systemic immune processes. Contracting skeletal muscles secrete cytokines (myokines), in particular interleukin-6 (IL-6), which mediates the metabolic changes that occur during exercise [56,57]. IL-6 release from muscles increases up to 100-fold during exercise contraction, resulting in an increased systemic production of anti-inflammatory cytokines, including interleukin-10 (IL-10) and interleukin-1 (IL-1) receptor antagonists, and decreased production of TNF-α and IL-1β [58,59]. Exercise-induced improvements in the inflammatory profile may also result from the modulation of intracellular signaling pathways and cellular functions mediated by NO and reactive oxygen species (ROS). The induction of anti-inflammatory defense mechanisms involves an increase in the expression levels of genes encoding antioxidant enzymes and heat shock proteins [60,61].

Although physical activity is effective in reducing the risk of many chronic diseases [62], acute training that is inappropriate in terms of intensity and duration for the capacity of the trainee can cause damage to muscle and connective tissue, typically manifested by the infiltration of inflammatory cells into the damaged tissue and the presence of pro-inflammatory cytokines in the peripheral blood [56]. The inflammatory response to exercise is attenuated when exercise is repeated at moderate intensity as the body adapts to the overload that occurs. Regular moderate exercise leads to a reduction in the baseline levels of circulating inflammatory markers as well as a reduction in the inflammatory response to intense exercise [61].

### 3.4. Equipotent Neuroprotective Effect of Steady Moderate and Intensive Physical Training

We compared the currently presented results, in which we used steady moderate exercise, to our previously published data [18], in which mice with induced Parkinsonism were subjected to high-intensity exercise. When subjected to steady moderate exercise, mice ran 360 m/day, whereas mice subjected to intensive exercise [18] ran 450 m daily. Importantly, the run velocity was constant (15 cm/s) during moderate exercise, while during intensive exercise [18], the velocity was variable (10–25 cm/s) and its mean value was 18.75 cm/s. There was a 25% difference between constant speed during moderate-intensity training and average speed during high-intensity training. Given the same daily training time (40 min), this also means a 25% difference in the distance run. Such a difference approximately corresponds to the difference between moderate and intensive exercise in human studies.

Comparing the results of the moderate-intensity training groups to those of the high-intensity training groups shows that the neuroprotection provided by the former type of training was at least as effective as that provided by the latter type of training. MPTP treatment in sedentary mice reduced the number of TH-ir neurons in the SNpc by 66%. Interestingly, the loss of dopaminergic neurons was less in the moderate-intensity training groups than in the high-intensity training mice (27 vs. 31% after early-onset training and 18 vs. 26% after late-onset training), although these differences were not significant (Appendix A).

BDNF levels in the midbrain, as well as the striatum, were non-significantly lower in untrained MPTP-treated mice compared to untrained and untreated control mice. Both moderate- and high-intensity training elevated the BDNF level in MPTP-untreated mice. In the midbrain, treatment with MPTP significantly potentiated the effects of both moderate- and high-intensity training as the BDNF level was significantly higher in MPTP-trained groups than in non-treated trained groups (Appendix A).

MPTP-treated sedentary mice had non-significantly reduced GDNF levels in the midbrain and significantly reduced GDNF levels in the striatum compared to untrained and trained controls. The effect of training and its intensity on GDNF levels was more pronounced in the striatum than in the midbrain. In the striatum, high-intensity training caused a significantly greater increase in GDNF in trained MPTP-untreated mice than moderate-intensity training. Similarly, high-intensity exercise caused a greater increase in GDNF levels than moderate exercise in MPTP-treated mice, but the difference was only present between the early-onset training groups (Appendix A).

It is also reasonable to suggest that the exercise-induced reduction in the inflammatory response may be the result of the activation of trophic factor-dependent signaling pathways triggered by an increased availability of neurotrophins [43]. For example, the anti-inflammatory activated microglia show increased expression of cytokines recognized as anti-inflammatory, such as IL-10, TGFβ, IGF-1, NGF, and BDNF [63,64]. Astrocytes, like microglia, also secrete anti-inflammatory agents into the environment, including neurotrophic factors (e.g., GDNF, BDNF, and mesencephalic astrocyte-derived neurotrophic factor (MANF)), which stimulate the survival and resuscitation of damaged dopaminergic neurons [65,66]. In addition, endogenous IL-1β has been shown to induce GDNF gene expression, synthesis, and secretion under in vitro conditions [67,68]. It is possible that exercise training in Parkinsonian MPTP mice induces an alternative neuroprotective activation of microglia rather than reducing pro-inflammatory glial activation.

On the other hand, it can be assumed that there is no increased pro-inflammatory proliferation and activation of glial cells as a result of prolonged exercise because neurotrophin-protected dopaminergic neurons do not degenerate and thus do not send signals that mobilize the inflammatory response system [7].

### 3.5. Importance of Exercise Intensity in Exercise-Induced Neuroprotection

The conclusion of a number of human studies is that only intensive exercise performed by PD patients slows motor deterioration [7,8], whereas there are not many publications confirming the beneficial effects of intensive exercise on the functionality of rescued dopaminergic neurons in the substantia nigra and the protection of nigrostriatal terminals in the putamen. Zigmond et al. [69] presented results obtained in three rhesus monkeys injected unilaterally with MPTP, with the other side of the brain serving as a control. One monkey was sedentary, the second performed treadmill running at 60% of the maximum heart rate (HRmax), and the third performed running at 80% HRmax. Running lasted 12 weeks before MPTP and 5 weeks after MPTP. The levels of TH, vesicular monoamine transporter 2 [VMAT2], and dopamine transporter (DAT) were undetectable in the caudate and putamen of the MPTP-treated hemisphere in the sedentary monkey; in the monkey running at 60% HRmax, only TH was barely detectable in this hemisphere. Running at 80% HRmax maintained TH, VMAT2, and DAT at levels similar to the control hemisphere.

The results of the study by de Laat et al. [70] bear some resemblance to the results described above in rhesus monkeys, in that the intensive exercise performed by PD patients improved the functionality of the remaining dopaminergic neurons in the substantia nigra and protected the nigrostriatal terminals in the putamen. Ten patients with mild and early PD exercised above the moderate intensity level, i.e., their heart rate was above 60–65% of their individual HRmax, often reaching 80% HRmax. This type of exercise increased DAT levels in the substantia nigra and putamen, as measured by PET imaging. This increase was found to be statistically significant when compared to the expected decrease previously measured by Delva et al. [71]. The authors also found that intense exercise reversed the expected decrease in neuromelanin concentration in the substantia nigra, as assessed by MRI. Neuromelanin is produced in the cytosol from dopamine and other catecholamines, and its concentration in dopaminergic neurons increases linearly with age in healthy subjects [72]. In PD patients, neuromelanin concentration decreases significantly in the early years of the disease; so, an increase, rather than a decrease, in neuromelanin concentration can be interpreted as a renormalization of dopaminergic neuron activity.

The study cited above only looked at the effects of vigorous exercise; so, the question arises as to whether only vigorous exercise can reverse the progression of PD in sedentary PD patients. An indirect answer to this question is provided by the study by Schenkman et al. [9]. The authors compared the change in Unified Parkinson’s Disease Rating Scale (UPDRS) motor score caused by intensive treadmill exercise (80–85% HRmax) and moderate-intensity treadmill exercise (60–65% HRmax) for six months, 3 days per week. The mean change in UPDRS motor score was 0.3 (slight deterioration) in the high-intensity group, 2.0 in the moderate-intensity group, and 3.2 in the control group. Based on these results, the authors accepted the hypothesis that high-intensity exercise was associated with a significant attenuation of motor decline and rejected the hypothesis that moderate-intensity exercise could produce such a result.

If, in fact, very intensive exercises have to be used to positively modify the course of PD, this type of intervention will remain restricted to a limited number of Parkinson’s patients. It may become increasingly difficult to recruit truly sedentary people with PD and those in more advanced stages of the disease to systematically participate in intensive exercise. It should also be borne in mind that, according to current knowledge, intensive exercise can only slow the progression of the disease such that even people who exercise intensively will progress to a more advanced stage of PD.

As mentioned in the Introduction section, people with PD may find it difficult or impossible to participate in vigorous exercise. Muscle weakness in PD is a primary symptom [11] and exacerbates the normal age-related loss of maximal isometric strength, making people with PD weaker than healthy people of similar age [73]. Both maximal voluntary contraction and the rate of force development are reduced in PD, which may lead to a decrease in force production [74]. Sympathetic denervation reduces cardiac contractility, which may affect exercise capacity in patients with PD [70]. Fatigue is a debilitating and persistent symptom commonly reported in PD. Physical activity is also impaired in PD due to increased fatigability. This increase may be caused by reduced cardiorespiratory fitness, resulting in reduced oxygen delivery and utilization and leading to increased reliance on anaerobic metabolism [13].

It is also important to remember that exercise puts a strain on the body, especially the cardiovascular system. For example, exercise increases blood pressure. The extent to which blood pressure rises during exercise is determined by the brain’s pressure-regulating center. The increase in blood pressure in response to the same exercise is specific to each individual, and a certain percentage of exercisers, including patients with PD, will experience an above-average increase in arterial pressure [75]. A high increase in arterial pressure during exercise has been shown to be associated with a higher risk of cardiac events [76]. Although a cause-and-effect relationship between an exaggerated pressor response to exercise and cardiac events has not been established, individuals who experience particularly large increases in arterial pressure during exercise are systematically exposed to high cardiovascular stress.

It should be also taken into consideration that acute training that is inappropriate in terms of intensity and duration for the capacity of the trainee can cause damage to muscle and connective tissue, typically manifested by the infiltration of inflammatory cells into the damaged tissue and the presence of pro-inflammatory cytokines in the peripheral blood [56]. The inflammatory response to exercise is attenuated when exercise is repeated at moderate intensity as the body adapts to the overload that occurs. Regular moderate exercise leads to a reduction in the baseline levels of circulating inflammatory markers, as well as a reduction in the inflammatory response to intense exercise [61]. However, it should also be considered that very intense exercise may increase the risk of common infections and the anti-inflammatory response due to a downregulation of pro-inflammatory cytokines, together with the stimulation of anti-inflammatory cytokines [77].

The currently presented results obtained with steady moderate training and previously presented results from high-intensity training [18] show that increasing the intensity of exercise above the level applied in this study did not enhance the neuroprotective effect of aerobic training in this particular mouse model of PD induced by a series of MPTP injections. This suggests that there is a kind of saturation where the effect (% of neurons saved) has reached its highest possible value and further increases in exercise intensity do not produce a further increase in neuroprotection. Translating this finding into clinical practice means that it may not be rational to strive for maximum intensity of exercise as this may not enhance neuroprotection but may be detrimental and lead to the exclusion of many, if not the majority of, PD patients.

We think it is likely that the relationship between the intensity of exercise and neuroprotection could be a sigmoidal one. In such a case, in addition to saturation, which occurs when exercise intensity is high, there is also a sub-threshold range where this intensity is low and its change within this range has no effect on neuroprotection. The range between threshold and saturation is the range in which a change in exercise intensity will produce a change in neuroprotection. This range can be quite narrow, i.e., if the intensity of exercise falls below the level that still confers maximum neuroprotection, one would observe a disproportionate increase in neuronal loss. This seems to be the case in the study by Gerecke et al. [17]. Bearing in mind that they studied the neuroprotective effect of exercise applied before MPTP intoxication, the loss of dopaminergic neurons in the SNpc was 9.2% after a full distance run, while a 2/3 full distance run resulted in a loss of 30.7%, and a 1/3 full distance run resulted in a loss of 35.6% of neurons, which is close to the 41.5% loss in sedentary mice.

## 4. Materials and Methods

### 4.1. Experimental Groups: Drugs and Treatments

In the present study, we compared changes in the dopaminergic system, expression levels of trophic factors, and inflammatory processes under moderate-intensity training to those observed in a previous study in which mice with induced Parkinsonism were subjected to high-intensity training. The experiments were performed on cohorts of mice from the same generations of the breeding program, under the same conditions, by the same experimenter, and using the same methods of tissue processing and evaluation.

All procedures used in this study were approved by the First Local Ethics Committee for Animal Experiments in Warsaw and were performed in accordance with the Polish Law on the Protection of Animals and the National Institute of Health’s Guide for the Care and Use of Laboratory Animals (Publication No. 85-23, revised 1985) and the European Union Council Directive (63/2010/EU). Studies were performed in 12-week-old male C57BL/6J mice, purchased from the Medical University of Bialystok (Poland) and delivered to the Nencki Institute of Experimental Biology 1 month before the experiments. In this study, 35 control (vehicle-treated) and 52 mice treated with MPTP were used. Mice were kept on a 12:12 h light/dark cycle, with temperature and humidity maintained at 23 ± 1 °C and 55 ± 5%, respectively; food and water were freely available. With regards to the chronic MPTP administration protocol, mice were given 10 subcutaneous doses of MPTP 12.5 mg/kg dissolved in saline (Santa Cruz Biotechnology, San Diego, CA, USA, Cat No sc-206178; Axon Medchem, Groningen, The Netherlands, Cat No 1075) in combination with the intraperitoneal application of probenecid (250 mg/kg dissolved in dimethyl sulfoxide (DMSO); Sigma-Aldrich, St. Louis, MO, USA, Cat No P8761) injections for 5 weeks. The purpose of probenecid administration was to inhibit the rapid excretion and disintegration of MPTP and its metabolites from the brain and kidney; probenecid alone has no significant neurotoxic effect, but in combination, it potentiates the neurotoxicity of MPTP [78]. Control mice received saline and DMSO injections only. The mice were thus divided as follows: 19 mice in a control group (C), 16 mice in a control treadmill training group (C-T), 19 mice in an MPTP-injected group (MPTP), 17 mice in an MPTP-injected trained group that started treadmill training 1 week before the induction of Parkinsonism (MPTP-TE, i.e., MPTP + early-onset treadmill training), and 16 mice in an MPTP-injected trained group that started treadmill training immediately after the induction of Parkinsonism (MPTP-TL, i.e., MPTP + late-onset treadmill training) (Figure 6). The treadmill training lasted 40 min/day at 15 cm/s (360 m/day) and was performed 5 days/week for 10 weeks.

### 4.2. Animal Sacrifice and Brain Tissue Collection

Animals were initially induced into a sedative state with a mixture of ketamine (75 mg/kg; i.p.) and medetomidine (0.5 mg/kg; i.p.) and then deeply anesthetized (loss of corneal reflexes) by the intraperitoneal administration of Vetbutal at a dose of 1 mL/kg body weight. Intra-aortic perfusion was started when there was no reflex to skin stimulation. The tissues were rinsed with cold (4 °C) 0.1 M phosphate-buffered saline (PBS) containing 5 IU of heparin per 1 mL of buffer (50 mL within 5 min) and fixed with 4% paraformaldehyde in 0.1 M PBS (50 mL within 5 min). Tissues were cryoprotected with 5% glycerol and 2% DMSO in 0.1M PBS (50 mL within 5 min). Fluids were administered through a cannula connected to a peristaltic pump. After perfusion, the brains were removed, placed in fixative for 1 h, and then immersed in 10% glycerol and 2% DMSO (24 h) for cryoprotection, followed by 20% glycerol and 2% DMSO (24 h). The brains were kept at −80 °C until used for staining. Mouse brains were delivered on dry ice and cooled in a cryostat (Leica CM1850, Leica Bioststems, Deer Park, IL, USA) at −27 °C for 45 min. Frozen brains were fixed to the cryostat stage with a tissue freezing medium and sectioned coronally at 40 μm thickness.

### 4.3. Staining

#### 4.3.1. Nissl Staining

Sections were mounted in 0.03 M PB on gelled slides and dried at room temperature (RT) for several days. Sections were dewaxed by incubation for 3 min in the following ethanol solutions: 96%, 80%, and 70% acidified with acetic acid. This was followed by rinsing with distilled water and staining with 0.1% cresyl violet in acetate buffer for 15 min. After rinsing with distilled water, differentiation was carried out in a 70% ethanol solution with 10% acetic acid until no excess dye was present. The sections were then dehydrated in increasing concentrations of alcohol (one change of 80%, two changes of 96%, two changes of absolute alcohol). Next, the sections were incubated twice in the mixture of absolute alcohol and xylene (mixture I: 2 parts of absolute alcohol for 2 parts of xylene and mixture II: 1 part of absolute alcohol for 2 parts of xylene) and in 3 xylene solutions (3 min in each solution). The sections were finally mounted in a DePeX medium. Nissel-stained sections were used as reference labels for the identification of the brain structures that were to be analyzed.

#### 4.3.2. Immunohistochemistry for TH, DRD2, BDNF, and GDNF

For each individual antibody, all brain sections were processed simultaneously in the same reagents during the same labeling reaction. The sections were rinsed three times for 5 min with 0.1 M PBS. The sections were then incubated with 1% H_2_O_2_ solution in 0.1 M PBS to block the activity of endogenous peroxidases. The tissue material was then rinsed three times with 0.1 M PBS and incubated with 5% normal goat serum (NGS; Vector Laboratories, Inc., Newark, CA, USA) in 0.1 M PBS containing 0.3% TritonX-100 for one hour at RT. This was followed by incubation with primary antibodies in a solution containing 5% NGS, 1% bovine serum albumin (BSA), and 0.3% TritonX100 in 0.1 M PBS for one hour at RT and then overnight at 4 °C with continuous shaking. The following primary antibodies were used: anti-dopamine D2 receptor (DRD2, Invitrogen, Waltham, MA, USA; Cat #PA5-115142) at a dilution of 1:200, anti-glial cell line-derived neurotrophic factor (GDNF, Santa Cruz Biotechnology, Cat No sc-328) at a dilution of 1:500, anti-brain-derived neurotrophic factor (BDNF, Santa Cruz Biotechnology, Cat No sc-20981) at a dilution of 1:500, and anti-tyrosine hydroxylase (TH, MERCK, Darmstadt, Germany: Cat No AB152) at a dilution of 1:1000. After three washes with 0.1 M PBS, brain sections were incubated for 1 h at RT with secondary antibody solution (Biotinylated Goat Anti-Rabbit IgG Antibody, Vector Laboratories, Cat No BA-1000) (5% NGS, 1% BSA, and 0.3% TritonX100 in 0.1 M PBS) at a dilution of 1:200. The formulations were rinsed three times with 0.1 M PBS and then incubated with the Vectastain ABC Kit (Vector Laboratories, Inc., Newark, CA, USA) for 1 h at RT. To visualize primary–secondary antibody complexes, sections were incubated in PBS containing 3,3′-diaminobenzidine tetrahydrochloride (DAB, Sigma-Aldrich, Merck KGaA, Darmstadt, Germany), H_2_O_2_, and NiSO_4_ at concentrations of 0.025%, 0.0125%, and 0.04%, respectively. Stained sections were mounted, air-dried, and cover-slipped. The specificity of the immunostaining was checked by running some slides through the entire procedure without the primary antibody (first control) or secondary antibody (second control). No specific staining was observed in any of the control slides.

#### 4.3.3. Immunohistochemistry for Iba-1b

Brain sections were rinsed in 0.1 M PBS for 3 × 5 min, incubated with 1% H_2_O_2_ solution in 0.1 M PBS, and blocked with 5% normal rabbit serum (NRS; Vector Laboratories Inc., Newark, CA, USA) in 0.1 M PBS containing 0.3% TritonX100 for one hour at RT. The sections were then incubated with primary antibody against ionized calcium-binding adaptor molecule 1 (Iba-1b, Abcam Limited, Cambridge, UK; Cat No ab5076) diluted (1:500) in 0.1 M PBS containing 5% NRS, 1% BSA, and 0.3% TritonX100 for 1 h at RT and then overnight at 4 °C with continuous shaking. This was followed by 3 × 5 min PBS washes and one hour of incubation at RT with the solution of rabbit anti-goat IgG and horseradish peroxidase (HRP) conjugate. The secondary antibody was diluted (1:500) in 0.1 M PBS containing 5% NRS, 1% BSA, and 0.3% TritonX100. DAB was used as a chromogen to visualize the reaction. The stained sections were mounted, air-dried, and cover-slipped. Controls for immunohistochemistry were performed as described above.

#### 4.3.4. Immunofluorescence Staining for GFAP and CD11b

Free-floating sections were rinsed three times in 0.1 M PBS, incubated in 1% H_2_O_2_ solution in 0.1 M PBS for 30 min, and in 5% NGS solution (Vector Laboratories) in 0.1 M PBS with 0.3% TritonX-100 for 60 min. The next step was overnight incubation with anti-glial fibrillary acidic protein (GFAP, dilution 1:1000, Dako, Glostrup, Denmark) and anti-integrin subunit alpha M (CD11b, dilution 1:200, Bio-Rad Laboratories, Inc., Hercules, CA, USA) antibody solution in 0.1 M PBS with 0.3% TritonX100, 5% NGS, and 1% BSA. The next day, the sections were rinsed with 0.1 M PBS and transferred to a mixture of secondary antibodies (1:1000, ThermoFisher Scientific Inc., Waltham, MA USA for both GFAP and CD11b). Incubation with fluorescent antibodies was performed in the dark to prevent photobleaching. The sections were mounted on slides and cover-slipped using UltraCruz^®^ Aqueous Mounting Medium with DAPI (Santa Cruz Biotechnology, Inc., Dallas, TX, USA). The details for primary and secondary antibodies used are given in Appendix A.

#### 4.3.5. Enzyme-Linked Immunosorbent Assay for BDNF, GDNF, and GFAP

Five mice from each experimental group were decapitated, the brains were quickly removed, and the striatum (ST) and midbrain, including the substantia nigra (SN), were collected and stored at −80 °C until further use. Tissue samples were weighed and homogenized in 20 volumes of ice-cold homogenization buffer against wet tissue weight. After centrifugation, the resulting supernatants were used in an enzyme-linked immunosorbent assay (ELISA) to quantify striatal and midbrain levels of BDNF (ChemiKine Brain Derived Neurotrophic Factor, Sandwich ELISA Kit Cat. No. CYT306, Merck, Darmstadt, Germany), GDNF (Mouse GDNF/Glial Cell Line-Derived Neurotrophic Factor ELISA Kit, Cat No EIAab/e0043m, Filgen Inc., Nagoja, Japan), and GFAP (NS830, Merck Millipore, Darmstadt, Germany). All steps of the assay were performed according to the manufacturer’s recommendations.

### 4.4. Data Analyses

#### 4.4.1. Quantification of the Number of Dopaminergic Neurons in SNpc

The anterior to posterior extent of the SN was determined using the Paxinos and Franklin [79] mouse brain atlas. TH-immunoreactive neurons in the substantia nigra pars compacta (SNpc) (−3.15 to −3.51 posterior to Bregma) were imaged using a Nikon Eclipse Ni-E microscope (Nikon Instruments, Inc., Tokyo, Japan). Brain sections were selected for analysis so that the coronal cross-section through the substantia nigra corresponded to the level of the greatest dopaminergic cell distribution (i.e., between 400 and 700 microns from the rostral to caudal development of the SNpc) [80]. The boundaries of the region of interest (ROI), in this case, the area of the TH-ir neurons and their dendrites, were delineated using the X–Y plotting system of the NIS software (Nikon Instruments, Inc., Tokyo, Japan, https://www.microscope.healthcare.nikon.com/en_EU/products/software/nis-elements), which also measured the square area (in mm^2^) of the ROI. TH-immunoreactive neurons were counted under 100× magnification. We only counted neurons in which the neuronal bodies were immunoreactive for TH and the nucleus and proximal segment of one or two dendrites were clearly visible. Afterward, the mean packing density (mPD) of dopaminergic neurons was calculated as a function of the rostrocaudal level and location within the SNpc by using the determined number of cells and the square area of the ROI in each section analyzed. The following equation was used:$$mPD=\frac{\sum_{i=1}^{n}N_{i}}{\sum_{i=1}^{n}SA_{i}}\;$$
where mPD—mean packing density [mm^−2^]; N_i_—number of neurons counted in the i-TH section (positively stained against tyrosine hydroxylase); and SA_i_—square area of the i-TH analyzed ROI [mm^2^]. The mean number of TH-ir neurons found in each experimental group was expressed as the mean packing density [number of cells per one square millimeter].

#### 4.4.2. Densitometric Analysis of the Immunostaining of TH and DRD2 in the Striatum

DAB-stained sections were analyzed using a brightfield microscope (Zeiss, AXIO IMAGER.M1, Carl Zeiss Industrielle Messtechnik GmbH, Oberkochen, Germany). Fiji-ImageJ (v. 1.53t, https://fiji.sc/) software was used for quantitative analysis. Relative optical density (OD) was measured in ROI, i.e., the striatum. The digitization of the SNpc microscope images was carried out in a single microscopy session, and the OD measurements were carried out at the same time. In the first step, the observer delineated the ROI. In the second step, OD, i.e., the average level of greyness in the ROI, was calculated using ImageJ software. Mathematically, OD was the sum of the grey values of all pixels within the ROI divided by the number of these pixels. The threshold of OD was then determined by the observer. ImageJ allows all pixels with an OD below the threshold to be marked in red. In fact, it is possible to find a threshold value such that only the immunospecifically stained tissue is marked red. In order to find a suitable threshold value, the observer had to gradually change its value until all immunolabelled elements within the ROI became red. This procedure was performed once for the selected reference image. This threshold was then used to analyze all the other images. Using the threshold, a new measure was calculated: %Area. Specific OD was calculated as the mean OD of all pixels within the ROI with an OD below the threshold. %Area was calculated as the ratio of the number of pixels within the ROI with an OD below the threshold to the number of all pixels within that area.

#### 4.4.3. Quantification of the Levels of BDNF, GDNF, and GFAP in the Striatum and Midbrain

The optical density of the standards and samples processed in the ELISA test was determined using a microplate reader (ThermoLabsystem, Thermo Fisher Scientific Inc., Waltham, MA, USA) set at a wavelength of 450 nm, and the concentration of the tested samples was read from the standard curve and expressed as pg/mL for BDNF and GDNF and ng/mL for GFAP.

#### 4.4.4. Statistical Analysis

All datasets are expressed as group mean ± SD. A comparison between experimental groups was carried out with the use of a one-way analysis of variance (ANOVA) followed by a post hoc comparison with the Newman–Keuls test using STATISTICA 12 software (StatSoft Polska, Krakow, Poland, https://www.statsoft.pl/). The differences between groups were considered statistically significant for *p* ≤ 0.05.

## 5. Conclusions

Our previously published results [18] and those presented in this article show that there is a threshold of exercise intensity above which increasing exercise intensity no longer results in increased neuroprotection. Consistent with this finding, the increases in the neurotrophins BDNF and GDNF and the attenuation of inflammatory processes induced by steady moderate-intensity exercise were similar to those induced by high-intensity exercise. Future research should use a wide range of exercise intensities to determine how changes in exercise intensity affect neuroprotective efficacy. By examining changes in different processes and phenomena in relation to changes in exercise intensity, key determinants of exercise-induced neuroprotection may be identified.

## Figures and Tables

**Figure 1 ijms-26-01146-f001:**
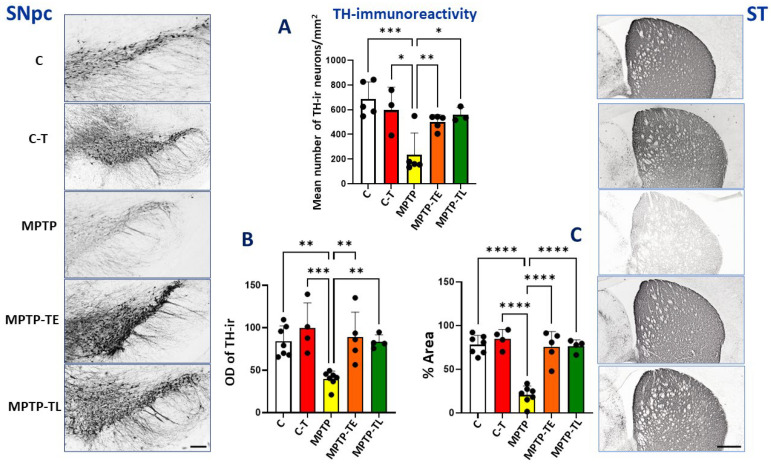
Tyrosine hydroxylase (TH) immunohistochemistry in the substantia nigra pars compacta (SNpc) and striatum (ST) of control mice (C, n = 5 in SNpc and n = 7 in ST), trained control mice (C-T, n = 3 in SNpc and n = 4 in ST), untrained mice with induced Parkinsonism (MPTP, n = 5 in SNpc and n = 7 in ST), mice with induced Parkinsonism that underwent early-onset training (MPTP-TE, n = 5 in SNpc and n = 5 in ST), and mice with induced Parkinsonism that underwent late-onset training (MPTP-TL, n = 3 in SNpc and n = 4 in ST). Representative images of the TH immunohistochemical staining in SNpc (SNpc left panel) and in the striatum (ST right panel). Magnification of microphotographs by 100 times for SNpc and 40 times for the striatum. (**A**) Mean number of TH-immunoreactive neurons in SNpc, (**B**) optical density (OD) of TH-immunoreactivity, and (**C**) % of immunoreactive area (%Area) in the dorsal striatum. Values are presented as the mean ± SD with scatter plots of individual data. (**A**) For TH-immunoreactive neurons in the SNpc, a one-way ANOVA confirmed a significant main effect of Group (F (4, 16) = 7.623; *p* = 0.0012). MPTP mice presented strongly reduced TH labeling relative to both control groups. While the mean number of the TH-ir neurons was reduced in MPTP relative to control groups, this reduction was restrained when physical exercise was administered. Both early-onset and late-onset treadmill training favorably affected the preservation of the dopaminergic phenotype of neurons in the SNpc. (**B**) For TH labeling in the striatum, statistical analysis yielded the main effect of Group for OD (F (4, 22) = 8.225; *p* = 0.0003) and (**C**) for %Area (F (4, 22) = 31.93; *p* < 0.0001). Neurotoxin administration reduced the density of dopaminergic afferents in the striatum, while exercise counteracted this reduction, regardless of when it was applied. Statistical significance between the groups was calculated using the Newman–Keuls test (* *p* < 0.05, ** *p* < 0.01, *** *p* < 0.001, and **** *p* < 0.0001). Scale bars: 100 µm for SNpc and 500 µm for ST.

**Figure 2 ijms-26-01146-f002:**
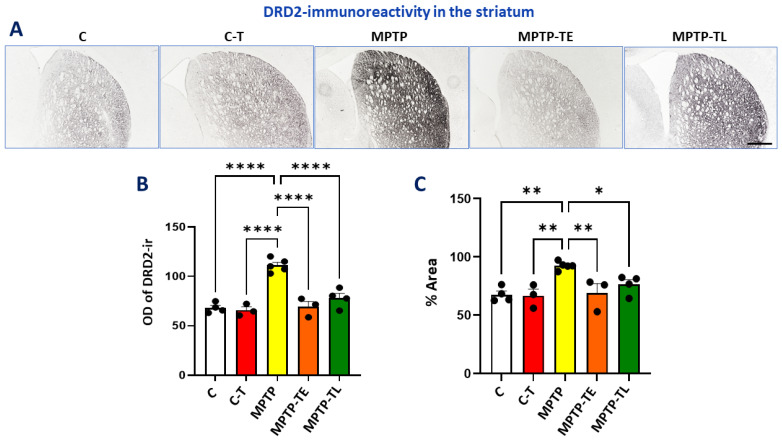
Dopamine receptor D2 (DRD2) immunohistochemistry in the striatum of control mice (C, n = 4), trained control mice (C-T, n = 3), untrained mice with induced Parkinsonism (MPTP, n = 5), mice with induced Parkinsonism that underwent early-onset training (MPTP-TE, n = 3), and mice with induced Parkinsonism that underwent late-onset training (MPTP-TL, n = 4). (**A**) Representative images of the DRD2 immunohistochemical staining in the striatum. Magnification of microphotographs by 40 times. (**B**) Optical density (OD) of DRD2-immunoreactivity and (**C**) % of the immunoreactive area (%Area) in the dorsal striatum. Values are presented as the mean ± SD with scatter plots of individual data. (**B**) For DRD2 labeling in the striatum, statistical analysis yielded the main effects of Group for OD (F (4, 14) = 28.25; *p* < 0.0001) and (**C**) for %Area (F (4, 14) = 7.409; *p* = 0.0020). Neurotoxin administration increased striatal DRD2 expression in MPTP mice (relative to C and to C-T), while physical training normalized DRD2 levels, as measured by OD and %Area. Statistical significance between the groups was calculated using the Newman–Keuls test (* *p* < 0.05, ** *p* < 0.01, and **** *p* < 0.0001). Scale bar: 500 µm.

**Figure 3 ijms-26-01146-f003:**
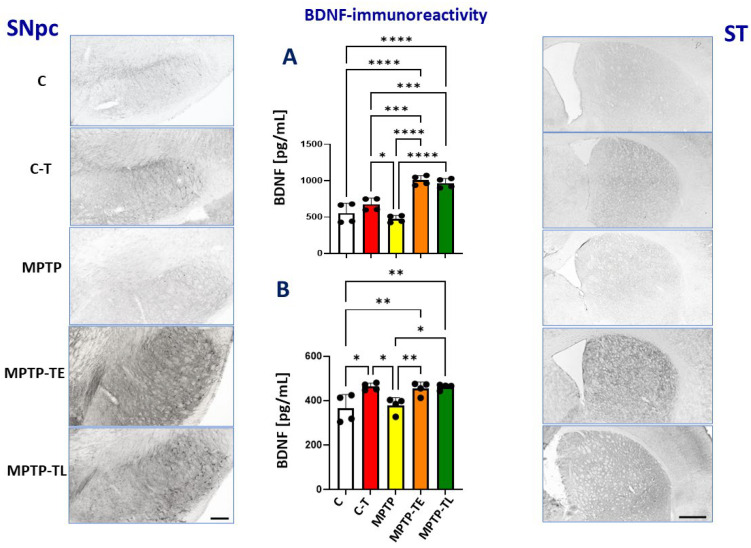
Brain-derived neurotrophic factor (BDNF) immunohistochemistry in the substantia nigra pars compacta (SNpc) and striatum (ST) of control mice (C, n = 4), trained control mice (C-T, n = 4), untrained mice with induced Parkinsonism (MPTP, n = 4), mice with induced Parkinsonism that underwent early-onset training (MPTP-TE, n = 4), and mice with induced Parkinsonism that underwent late-onset training (MPTP-TL, n = 4). Representative images of the BDNF immunohistochemical staining in SNpc (SNpc left panel) and the striatum (ST right panel). Magnification of microphotographs by 100 times for SNpc and 40 times for the striatum. Quantitative analysis of the levels of BDNF by ELISA in the midbrain (**A**) and the striatum (**B**). Values are presented as the mean ± SD with scatter plots of individual data. (**A**) Quantitative analysis of the BDNF level in the midbrain with one-way ANOVA confirmed a significant main effect of Group (F (4, 15) = 31.49; *p* < 0.0001). Untrained MPTP mice did not show a strongly reduced BDNF level compared to the control group (group C), whereas BDNF levels were elevated when physical exercise was administered in the control (C-T) and reached even higher levels when training was combined with MPTP treatment (MPTP-TE and MPTP-TL). (**B**) Quantitative analysis of the BDNF level in the striatum with one-way ANOVA yielded the main effects of Group (F (4, 15) = 7.357; *p* = 0.0017). Similarly, as observed in the midbrain, untrained MPTP mice did not show a strongly reduced BDNF level compared to control group (group C), whereas BDNF levels were similarly elevated when physical exercise was administered in the control (C-T) and when training was combined with MPTP treatment (MPTP-TE and MPTP-TL). Statistical significance between the groups was calculated using the Newman–Keuls test (* *p* < 0.05, ** *p* < 0.01, *** *p* < 0.001, and **** *p* < 0.0001). Scale bars: 100 µm for SNpc and 500 µm for ST.

**Figure 4 ijms-26-01146-f004:**
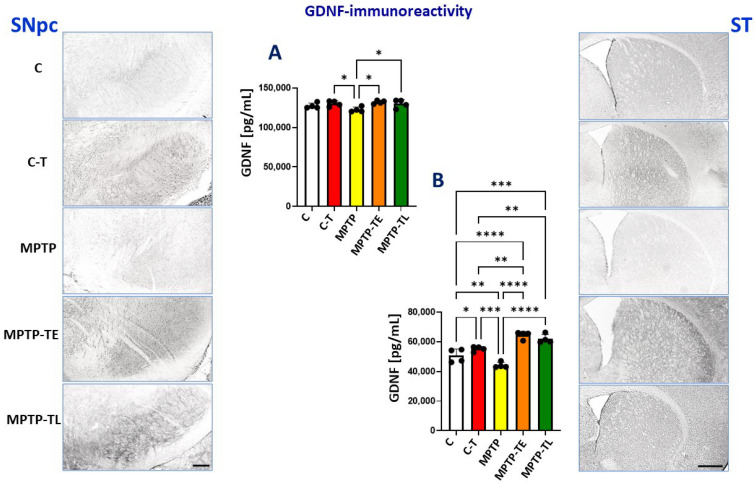
Glial-derived neurotrophic factor (GDNF) immunohistochemistry in the substantia nigra pars compacta (SNpc) and striatum (ST) of control mice (C, n = 4), trained control mice (C-T, n = 4), untrained mice with induced Parkinsonism (MPTP, n = 4), mice with induced Parkinsonism that underwent early-onset training (MPTP-TE, n = 4), and mice with induced Parkinsonism that underwent late-onset training (MPTP-TL, n = 4). Representative images of the GDNF immunohistochemical staining in SNpc (SNpc left panel) and the striatum (ST right panel). Magnification of microphotographs by 100 times for SNpc and 40 times for the striatum. Quantitative analysis of the levels of GDNF by ELISA in the midbrain (**A**) and the striatum (**B**). Values are presented as the mean ± SD with scatter plots of individual data. (**A**) Quantitative analysis of the GDNF level in the midbrain with one-way ANOVA showed a significant main effect of Group (F (4, 15) = 3.976; *p* = 0.0215). MPTP mice showed a non-significantly decreased GDNF level compared to the control group (group C). GDNF levels were not reduced in trained MPTP mice when compared to control groups (C, C-T). (**B**) Quantitative analysis of the GDNF level in the striatum with one-way ANOVA yielded the main effects of Group (F (4, 15) = 31.76; *p* < 0.0001). Unlike in the SNpc, untrained MPTP mice showed significantly decreased GDNF levels compared to control groups, whereas GDNF levels were elevated when physical exercise was administered in both control and MPTP-treated mice. The pro-trophic effect of exercise was particularly pronounced in mice receiving MPTP, both in the early-onset and late-onset training groups. Statistical significance between the groups was calculated using the Newman–Keuls test (* *p* < 0.05, ** *p* < 0.01, *** *p* < 0.001, and **** *p* < 0.0001). Scale bars: 100 µm for SNpc and 500 µm for ST.

**Figure 5 ijms-26-01146-f005:**
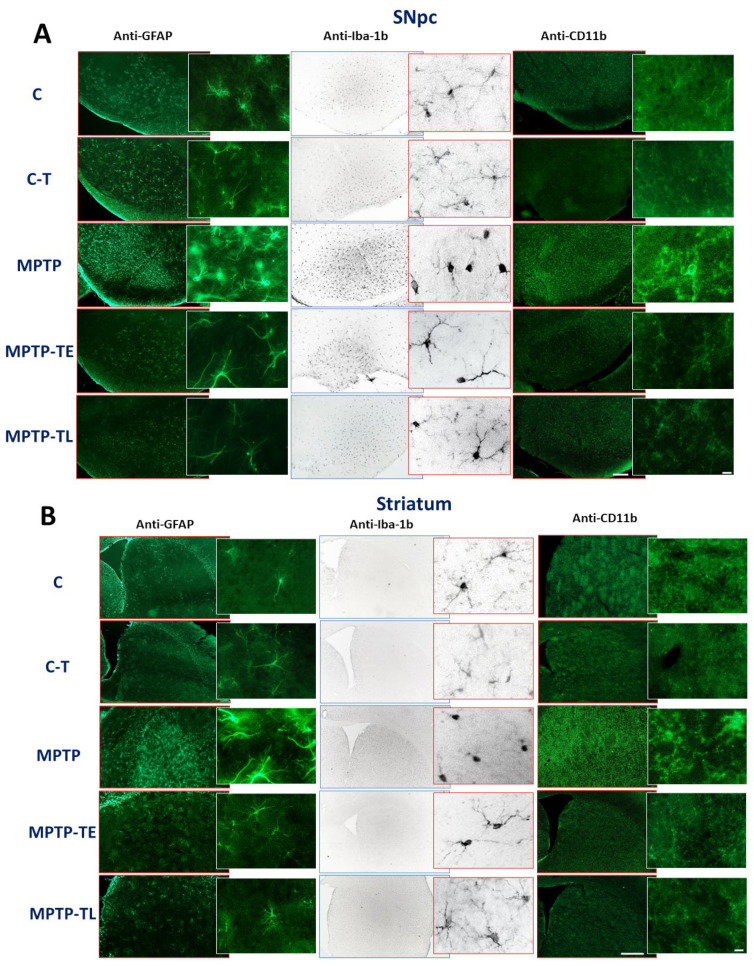
Immunohistochemical staining against glial fibrillary acid protein (GFAP), ionized calcium-binding adaptor molecule 1 (Iba-1b), and integrin CD 11b in the (**A**) substantia nigra pars compacta (SNpc) and (**B**) striatum (ST) of control mice (C), trained control mice (C-T), untrained mice with induced Parkinsonism (MPTP), mice with induced Parkinsonism that underwent early-onset training (MPTP-TE), and mice with induced Parkinsonism that underwent late-onset training (MPTP-TL). Representative images of GFAP, Iba-1b, and CD11b immunohistochemical staining in SNpc and ST. Scale bars: 100 µm for SNpc, 500 µm for ST, and 10 µm for insertions.

**Figure 6 ijms-26-01146-f006:**
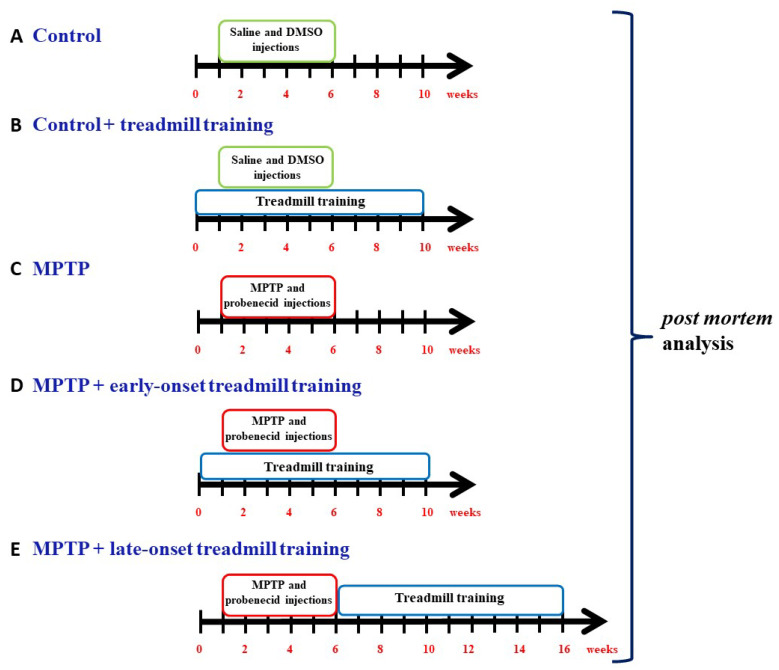
Schematic diagram of the study design. A detailed description of the research protocol can be found in the Methods section. MPTP—1-N-Methyl-4-phenyl-1,2,3,6-tetrahydropyridine, a precursor to the neurotoxin 1-Methyl-4-phenylpyridinium (MPP^+^) and DMSO—dimethyl sulfoxide.

## Data Availability

The data presented in this study are available upon request from the corresponding author. Please refer to the Appendix A for additional data supporting the reported results.

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
