# Peer review of "Steady Moderate Exercise Confers Resilience Against Neurodegeneration and Neuroinflammation in a Mouse Model of Parkinson’s Disease"

_ijms, 2025, doi:10.3390/ijms26031146_

Round 1
Reviewer 1 Report
Comments and Suggestions for Authors
The manuscript explores a pertinent topic in Parkinson's disease (PD) research, focusing on moderate exercise as a potential therapeutic strategy to alleviate neurodegeneration and neuroinflammation. This will undoubtedly contribute to advancing knowledge in the field. However, there are some major and minor issues that require revision, which I have outlined below.
* The abstract could benefit from more specific details about the study's findings.
* In the Introduction, consider condensing lines 54-65 that describe the results of studies related to the intensity and types of exercise and their outcomes for PD.
* Consider rephrasing lines 71-74, as there are studies that have been conducted involving PD patients to understand the underlying mechanisms of exercise against the disease. The research involving human subjects is mentioned in Lines 90-91.
* The sentence "The experiments were performed under the same conditions, by the same experimenter, using the same methods of tissue processing and evaluation" (Lines 99-100) could be relocated to the Methods section.
* The Methods are detailed, which is beneficial for future reference. Please include the dose of Vetbutal (Line 138).
* Consider moving Table 1 (antibodies) to the supplementary material. This would help to reduce the current length of the paper.
* Improve the quality of immunohistochemistry figures and include scale bars in each figure. Also, include the number of animals in the figure legends.
* A major point of this paper is that the Discussion section is extensive and requires significant reduction. There are not many results in the study; however, there is repetition in describing the results that could be condensed. The authors also discuss aspects that were not addressed in the study. Please, focus only on what is relevant to the current study. Additionally, it would be useful to mention some possible limitations of the work.
* Define all abbreviations in the text and use them consistently. Additionally, improve English for clarity.
Author Response
Responses to the comments made by the Reviewer #1
Dear Reviewer,
We would like to express our sincere gratitude for your valuable feedback and constructive comments on our manuscript. Your suggestions have been very helpful in improving the quality and clarity of our work. We have carefully considered each issue and made the necessary revisions, which are highlighted in the main text. Below you will find detailed responses to each of your comments.
Yours sincerely on behalf of all the authors,
Grazyna Niewiadomska
Reviewer comment #1.
The abstract could benefit from more specific details about the study's findings.
Response: We agree with the Reviewer that the abstract would have been more informative if it had included more information about the research results. We have therefore expanded its content; please compare the changes highlighted in yellow, lines 26 to 40.
Reviewer comment #2.
In the Introduction, consider condensing lines 54-65 that describe the results of studies related to the intensity and types of exercise and their outcomes for PD.
Response: We have followed the Reviewer's suggestion and rewritten the indicated section of the Introduction. This has resulted in the removal of the former item number 9 from the References.
Reviewer comment #3.
Consider rephrasing lines 71-74, as there are studies that have been conducted involving PD patients to understand the underlying mechanisms of exercise against the disease. The research involving human subjects is mentioned in Lines 90-91.
Response: The indicated paragraph has been substantially rewritten to include the Reviewer's suggested information on the mechanisms of beneficial effects of physical activity on disease progression in Parkinson's disease patients; please compare the changes highlighted in yellow, lines 69 to 89.
Reviewer comment #4.
The sentence "The experiments were performed under the same conditions, by the same experimenter, using the same methods of tissue processing and evaluation" (Lines 99-100) could be relocated to the Methods.
Response: Following the validt comment made by the Reviewer, the mentioned sentence has been moved to Methods section; please compare the changes highlighted in yellow, lines 131 to 136.
Reviewer comment #5.
The Methods are detailed, which is beneficial for future reference. Please include the dose of Vetbutal (Line 138).
Response: Vetbutal dose information was included in the Methods section, lines 169 to 172.
Reviewer comment #6.
Consider moving Table 1 (antibodies) to the supplementary material. This would help to reduce the current length of the paper.
Response: Table 1 has been moved to the Supplementary Material, as suggested by the Reviewer.
Reviewer comment #7.
Improve the quality of immunohistochemistry figures and include scale bars in each figure. Also, include the number of animals in the figure legends.
Response: The scale bars have been included in the figures. Information on the size of the experimental groups has been added to the description of the figures.
We have tried to include the best possible microscopic images of the immunohistochemical studies. Some discomfort may be caused by the fact that these are labels of very different structures. The large TH-ir neurons and GFAP-ir astrocytes are very clearly labelled. In contrast, it is more difficult to get a perfect image of small microglial cells, especially those labelled with CD11b. This protein is preferentially located in the cell membrane and may also be partially present in the extracellular space, making the image of its labelling less sharp. Similarly, difficulties may be associated with the dispersive immunolabelling of neurotrophins and the Th-ir and DRD2-ir neuropil in the striatum. It is our hope that the reviewer will take the above comments into account.
Reviewer comment #8.
A major point of this paper is that the Discussion section is extensive and requires significant reduction. There are not many results in the study; however, there is repetition in describing the results that could be condensed. The authors also discuss aspects that were not addressed in the study. Please, focus only on what is relevant to the current study. Additionally, it would be useful to mention some possible limitations of the work.
Response: Recognising that the Reviewer's comments were valid, we have made very substantial changes to the content of the Discussion. Large sections of the text have been removed, and others have been reworded. We would like to leave the last paragraph of the Discussion (4.5.) a little more elaborate (although also abbreviated), because it shows how important the intensity of physical exercise used as an adjunctive therapy in PD is in clinical management. Admittedly, our study is only an animal model study, but we believe it can provide important information on this issue.All changes have been marked in the new version of the manuscript, either as yellow underlining or as deletions.
Reviewer comment #9.
Define all abbreviations in the text and use them consistently. Additionally, improve English for clarity.
Response: There has been a revision of the definition of acronyms and a reorganisation of their use in the text of the manuscript. The manuscript has undergone another round of linguistic revision.
Reviewer 2 Report
Comments and Suggestions for Authors
The study investigates the neuroprotective effects of moderate aerobic exercise on a mouse model of Parkinson’s disease (PD) induced by chronic MPTP administration. Furthermore, the authors declare that moderate-intensity treadmill exercise provided a level of neuroprotection similar to high-intensity exercise, suggesting that exercise intensity may not need to be maximized to achieve benefits. The results show saturation in neuroprotection, meaning further intensity does not yield additional neuronal rescue. The findings are interesting.
However, there are some concerns raised:
1, the author just compared the moderate exercise in the current study with previous intensive exercise. The conclusion is not consolidated, as there are maybe some differences between current study and previous study. An intensive exercise group should be added in the current study and compare with moderate exercise mice.
2, it is well known that exercise can activate PGC-1a pathway to protect neurons. The authors should check PGC-1a expression levels before and after exercise.
Author Response
Responses to the comments made by the Reviewer #2
Dear Reviewer,
We would like to thank you for your insightful feedback on our manuscript. We have carefully considered each of your comments, and below are our detailed responses to your questions.
Kind regards on behalf of all the authors,
Grażyna Niewiadomska
Reviewer comment #1.
The author just compared the moderate exercise in the current study with previous intensive exercise. The conclusion is not consolidated, as there are maybe some differences between current study and previous study. An intensive exercise group should be added in the current study and compare with moderate exercise mice.
Response: The results presented in this article and in the previously published article are from a single study carried out on cohorts of mice from the same generations of the breeding programme, under the same conditions, by the same experimenter, using the same methods of tissue processing and evaluation. In view of the above, we believe that it is legitimate to compare the results obtained for the two types of training. In the previous publication, we refrained from publishing the results obtained from mice undergoing constant moderate-intensity exercise because they replicated almost exactly the results obtained from mice undergoing high-intensity exercise. We have now decided to publish this part of the results because we realised that the question of whether only high-intensity exercise confers neuroprotection in Parkinson's patients has recently become a clinical issue. There is a lot of evidence that moderate-intensity exercise, which is safer for patients, may also be effective in treating the disease.
Reviewer comment #2.
It is well known that exercise can activate PGC-1a pathway to protect neurons. The authors should check PGC-1a expression levels before and after exercise
Response: Undoubtedly, studying the peroxisome proliferator-activated receptor γ coactivator-1α (PGC-1α) pathway would shed light on the mechanisms underlying the relationship between exercise intensity and brain BDNF levels, especially in light of our observation that increasing exercise intensity did not affect midbrain BDNF levels. Muscle PGC-1α affects cerebral BDNF via the fibronectin type III domain-containing protein 5 (FNDC5)/irisin pathway, whereas in the brain PGC-1α affects BDNF levels as part of the sirtuin 1/ PGC-1α/FNDC5 pathway. Cerebral BDNF levels can also be influenced by non-PGC-1α factors such as beta-hydroxybutyrate, glycosylphosphatidylinositol-specific phospholipase D1 (Gpld1), fibroblast growth factor 21 or insulin-like growth factor 1 (IGF-1). It is also known that muscles involved in exercise can also be a source of increased BDNF levels in the brain. Derived from the periphery, BDNF has the ability to cross the BBB.
Which of these factors and pathways, and how they respond to exercise intensity, may help explain the relationship between exercise intensity and neuroprotection. However, the result of these and probably other interactions is a beneficial change in brain BDNF levels, so we believe that by reporting our observations we have provided important information that may relate to neuroprotection.
At present, we are unable to investigate PGC-1α levels due to a lack of tissue, which has been fully utilised in a number of biochemical stainings on both brain slices and brain tissue homogenates in immunoassays. However, we are grateful for the attention drawn to PGC-1α and its investigation will be considered in future projects. We recognise how important PGC-1α can be, and although we were not able to identify it in the current research, we highlight its importance in the Introduction (line 77).
Round 2
Reviewer 1 Report
Comments and Suggestions for Authors
The authors have sufficiently addressed my comments.
Author Response
Rreviewer's comment: The authors have sufficiently addressed my comments.
Response: We are pleased and gratified to learn that the revision of our manuscript meets the Reviewer's expectations.
We are very grateful to the Reviewer for his careful and kind assessment of our work, which has significantly improved its quality.
Reviewer 2 Report
Comments and Suggestions for Authors
The authors have extensively revised their article and addressed reviewer's comments. So I suggest accepting it now
Author Response
Reviewer's comment: The authors have extensively revised their article and addressed reviewer's comments. So I suggest accepting it now.
Rresponse: We would like to express our sincere thanks to the reviewer for his or her efforts in evaluating our manuscript, and we are very pleased that it has been recommended for publication.